# Ultrafast Microfluidic PCR Thermocycler for Nucleic Acid Amplification

**DOI:** 10.3390/mi14030658

**Published:** 2023-03-15

**Authors:** Yi-Quan An, Shao-Lei Huang, Bang-Chao Xi, Xiang-Lian Gong, Jun-Hao Ji, You Hu, Yi-Jie Ding, Dong-Xu Zhang, Sheng-Xiang Ge, Jun Zhang, Ning-Shao Xia

**Affiliations:** State Key Laboratory of Molecular Vaccinology and Molecular Diagnostics, National Institute of Diagnostics and Vaccine Development in Infectious Diseases, Department of Laboratory Medicine, School of Public Heath, Xiamen University, Xiamen 361102, China

**Keywords:** polymerase chain reaction, thermocycler, heat transfer modeling analysis, temperature overshoots, fast heating and cooling ramp rates

## Abstract

The polymerase chain reaction (PCR) is essential in nucleic acid amplification tests and is widely used in many applications such as infectious disease detection, tumor screening, and food safety testing; however, most PCR devices have inefficient heating and cooling ramp rates for the solution, which significantly limit their application in special scenarios such as hospital emergencies, airports, and customs. Here, we propose a temperature control strategy to significantly increase the ramp rates for the solution temperature by switching microfluidic chips between multiple temperature zones and excessively increasing the temperature difference between temperature zones and the solution; accordingly, we have designed an ultrafast thermocycler. The results showed that the ramp rates of the solution temperature are a linear function of temperature differences within a range, and a larger temperature difference would result in faster ramp rates. The maximum heating and cooling ramp rates of the 25 μL solution reached 24.12 °C/s and 25.28 °C/s, respectively, and the average ramp rate was 13.33 °C/s, 6–8 times higher than that of conventional commercial PCR devices. The thermocycler achieved 9 min (1 min pre-denaturation + 45 PCR cycles) ultrafast nucleic acid amplification, shortening the time by 92% compared to the conventional 120 min nucleic acid amplification, and has the potential to be used for rapid nucleic acid detection.

## 1. Introduction

Nucleic acid testing (NAT) is the gold standard for pathogenic microorganism detection because of its high sensitivity and good specificity [1]; it is used in many applications such as infectious disease detection, tumor screening, forensic identification, genomic programs, food safety testing, and environmental monitoring [2,3,4]. For example, the main criterion for identifying COVID-19 infection is the results of NAT [5,6]. Nucleic acid amplification (NAA), an important step in NAT, enables the mass amplification of nucleic acid molecules over a period of time. The polymerase chain reaction (PCR) is the most commonly used NAA technique in clinical practice and generally consists of pre-denaturation, denaturation, annealing, and extension stages; PCR instruments (also called thermocyclers) provide the appropriate temperature conditions for this process. However, current commercial PCR instruments require 1~3 h for testing, making it increasingly difficult to meet the demand for rapid clinical NAT and highlighting the need for rapid thermocyclers for scenarios requiring rapid NAT results such as airports, emergency care, and surgery.

A number of researchers have studied rapid thermocyclers in more depth, and miniaturized PCR instruments that can achieve faster temperature changes with low-heat-capacity reactors and microfluidics are becoming increasingly popular [7]. The heating and cooling ramp rates of the solution in these instruments range from 4 to 180 °C/s due to varying liquid volumes and heating/cooling methods. Product detection is commonly performed using agarose gel electrophoresis and fluorescence detection. The previous studies are summarized in Table 1.

Taken together, the current implementations of thermocyclers can be divided into three main types: thermal convection, fixed microchamber, and flow channel types [17]. The thermal convection type is used for NAA by inducing solution thermal convection between different regions in the reactor [18,19], which can allow faster temperature changes but the solution cannot maintain a stable and prolonged constant temperature, so the amplification efficiency is questionable [20]. The fixed microchamber type is where the solution is fixed in a tiny chamber and heated and cooled by the heaters. In recent years, new heating methods have also emerged for the fixed microchamber type, such as infrared heating and plasma–laser heating [12,21,22]. The detection sensitivity of the fixed microchamber type can be effectively guaranteed, but the traditional Peltier-based heating method cannot achieve rapid PCR because temperature changes take a long time; additionally, the new heating methods heat quickly but cool slowly, require expensive optical components such as light sources, lenses, and filters; and may have limited potential in clinical applications. The flow channel type is where the solution flows in the microchannel of the reactor through different thermostatic zones to obtain the desired temperature conditions, which enables rapid NAA [23,24] but increases the number of biomolecules adsorbed on the channel surface and inhibits PCR, resulting in reduced detection sensitivity; in addition, this type requires expensive external pumps and struggles to achieve quantitative detection.

The current thermocyclers have the following problems: (1) most devices have slow solution temperature ramp rates, with amplification times exceeding 30 min; (2) ultrafast PCR thermocyclers have small solution volumes (<10 μL), which may reduce sensitivity, and have limited potential in clinical applications [25]; (3) most studies only describe the designed systems and their performance, without mechanistically analyzing in depth the key factors affecting the ramp rates.

Therefore, it is of great value to develop an ultrafast thermocycler with clinical application potential by analyzing in depth the factors affecting the ramp rates. This study proposes a temperature-control strategy to significantly increase the solution ramp rates by switching reactors between multiple temperature zones and excessively increasing the temperature difference between the temperature zones and the solution; accordingly, we designed an ultrafast thermocycler, where a microfluidic chip containing the solution is cycled between three constant-temperature zones with special settings, which combines the advantages of the fixed microchamber and flow channel types of thermocyclers, and provides ultrafast, accurate, and stable temperature conditions for PCR. Firstly, we modeled and analyzed the key factors affecting the ramp rates of the solution based on heat-transfer theory and designed and built a test platform based on relevant computational data; secondly, we explored the variation in ramp rates using the temperature overshoot strategy; and finally, we conducted thermal cycling performance tests of the thermocycler and PCR assays of real biological samples (human cytomegalovirus, HCMV).

## 2. Materials and Methods

### 2.1. Mechanical Structure of Thermocycler

#### 2.1.1. Requirement Analysis and Design Scheme

To achieve rapid nucleic acid amplification detection, annealing and extension are achieved simultaneously by adjusting their required temperature to a suitable temperature when the target DNA fragment is short (<300 bp). Hence, only two temperatures were needed for PCR to allow the denaturation and annealing/extension of DNA, which is also used in this study [26,27]. PCR requires accurate and stable temperature conditions and a rapid change in the solution temperature. In the denaturation stage, if the temperature is too high, it will affect the polymerase activity and reduce the efficiency of NAA, and if the temperature is too low, it will lead to insufficient denaturation and most likely false negatives. In the annealing stage, if the temperature is too high, it will affect the binding of primers to the template, and if it is too low, it will lead to nonspecific amplification. The speed of NAA depends mainly on the ramp rates of the solution temperature [28,29,30]. In addition, the annealing/extension and denaturation process generally requires 30~45 cycles of repetition in order to meet the requirements of nucleic acid detection [31].

With reference to heat transfer and thermoelectric equivalence theory, the main factors affecting the ramp rates of the solution may be the temperature difference, thermal resistance, and heat capacity of the heated body. The specific modeling analysis is shown in Section 2.3.

According to the temperature requirements of PCR and the potential main factors affecting the ramp rates, this study proposes an ultrafast PCR thermocycler scheme with three temperature-zones (TZs) and a temperature control strategy to substantially increase the ramp rates by over-raising the temperature difference. The three TZs were maintained at different constant temperatures: temperature zone I (TZ I) was far below 55 °C (50 °C, 40 °C, or 30 °C), temperature zone II (TZ II) was 55 °C, and temperature zone III (TZ III) was far above 95 °C (100 °C, 110 °C, or 120 °C). The flat microfluidic chips containing the solution were cyclically switched in each TZ to achieve the required temperature change of 95 °C and 55 °C for PCR, and the scheme is shown in Figure 1.

#### 2.1.2. Microfluidic Chip Design and Fabrication

The microfluidic chips were the site of the PCR. Each polycarbonate (PC) microfluidic chip body contained a sample inlet and an air outlet which were connected by two microchannels to a PCR amplification chamber that could hold 25 μL. The rest of the chip contained two 0.05 mm thick polycarbonate films and two rubber plugs, as shown in Figure 2. The chip body was machined to shape on a CNC and the films were cut on a laser cutter; the chip body and films were bonded together through ultrasonic bonding (AD2000, Xiamen Strong Long Automation Technology Co., Ltd., Xiamen, China). It was important to note that the solder lines must be processed on the chip body, otherwise the films could not be bonded to it through ultrasonic bonding.

#### 2.1.3. Structure Composition

The three-dimensional structure of the thermocycler is shown in Figure 3, with overall dimensions of 140 mm × 130 mm × 70 mm and a mass of 1.5 kg. The structure consisted of a support mechanism, a motion mechanism, temperature zone units, and a heat dissipation module. The motion mechanism consisted of two parts: the stepper motor module which drove the microfluidic chip to move horizontally among the TZs, and the linear servo driver which drove the upper substrate to move vertically closely fitted and separated the heating part and the microfluidic chip, respectively.

Each TZ unit was a symmetrical structure on the top and bottom, composed of heat-generating pieces, heat-conducting pieces, heat-insulating pieces, and PT1000 RTDs; the three TZs were connected into one whole through the substrate, as shown in Figure 1. The heat-generating pieces generated heat according to the working power. The heat-conducting pieces were made of purple copper with a large coefficient of thermal mass and conductivity, and the PT1000 RTDs were installed inside them for real-time temperature feedback to accurately control the temperature. The fitting surface between the heat-generating pieces and the heat-conducting pieces as well as between the heat-conducting pieces and RTDs was coated with a 0.02 mm thick layer of silicone grease to reduce the contact thermal resistance between two contacting parts and facilitate heat transfer. The heat-insulating pieces were made of Bakelite with a low thermal conductivity coefficient and were properly hollowed out (air is the best heat insulator) to reduce the heat loss from the heat generators and also to weaken the thermal crosstalk from the high-temperature TZ to the low-temperature TZ. In addition, a cooling fan was installed in TZ I to prevent the temperature of TZ-I and TZ-II becoming passively higher than the set temperature due to heat transfer.

### 2.2. Electric Control System of Thermocycler

The electronic control module of the thermocycler consisted of a master control unit, a communication and control unit, a temperature feedback unit, a temperature drive unit, and a motor control unit, as shown in Figure 4. The master control unit was developed based on STM32 and communicated with the computer through the serial port to receive various control commands. The temperature feedback unit collected and converted the Pt1000 RTD signal through the MAX31865 converter to read the temperature data. Based on the real-time feedback temperature, the temperature drive unit controlled the pulse width modulation (PWM) duty cycle through the proportional/integral/differential (PID) algorithm, and then adjusted the input power of the heaters through the power amplifier circuit to control the constant temperature of each temperature zone. In addition, the motor control unit controlled the TMC5130 driver chip to control the stepper motor via SPI communication and the linear servo driver via the serial port.

### 2.3. Modeling and Solving the Thermocycler’s Critical Parameters

In order to rapidly change the solution temperature, mathematical models were then performed to analyze the key factors affecting the ramp rates of the solution based on heat-transfer theory, as well as to calculate the amount of heat generated by the heat generator and to select its type. These approaches are applicable not only to this device but also to other rapid thermocyclers.

#### 2.3.1. Modeling of the Temperature in the Microfluidic Chip and Solution

Within the concept of thermoelectric equivalence, the driving temperature difference of heat transfer, the thermal resistance and the heat capacity of the reactor and the solution affect the ramp rates of the solution [32,33]. Since the microfluidic chip we used was a thin, flat-shaped chamber and the sealed film was only 0.05 mm, the temperature response of the microfluidic chip and solution could be understood using a simplified quasi-lumped capacitance transient analysis according to the heat-transfer theory [34]. When the microfluidic chip was moved, it and the solution with an initial temperature *T*_rs_ were brought quickly into contact, at time *t*_0_, with the heat-conducting pieces with a temperature of *T*_c_; their temperature was:(1)$$T\left(t\right)=T_{c}+\left(T_{\mathrm{rs}}-T_{c}\right)e^{-\left(t-t_{0}\right)/\tau }$$
where *R* is thermal resistance, and *C* is the heat capacity of the microfluidic chip and the solution. Therefore, *T*(*t*) can be increased by decreasing *R* or *C* to make it infinitely close to *T*_c_. The thermal resistance consists of the contact thermal resistance between the heat-conducting pieces and the microfluidic chip as well as the thermal conductivity thermal resistance of the microfluidic chip. The contact thermal resistance is reduced by increasing the compression force between the two surfaces, and the thermal conductivity thermal resistance is reduced by selecting a thin film with high thermal conductivity. In addition, there is also contact thermal resistance between the heater and the thermal conductor, which could be minimized by increasing the compression force between the two surfaces and applying silicone grease.

The time for the temperature of the microfluidic chip and the solution to reach a specific temperature *T* is
(2)$$t=-RC\ln\frac{T-T_{c}}{\Delta T}+t_{0}$$

When the thermal resistance *R* and the heat capacity *C* of the microfluidic chip and solution are certain, the larger the temperature difference, the smaller the time *t*. Therefore, it is effective to enhance the ramp rates of the solution by increasing the temperature difference between the solution and the heat source in excess.

Since the contact thermal resistance can only be reduced to some extent, and the conductive thermal resistance is larger due to the lower thermal conductivity of the chamber membrane of the microfluidic chip, their effect on the rate enhancement is very limited. Therefore, when the thermal resistance is large, the ramp rates of the solution can be significantly increased by increasing the temperature difference, which is one of the focuses of this study.

#### 2.3.2. Modeling of Temperature in the Heat-Conducting Pieces

Since the heat of the solution comes from or is released to the heat-conducting pieces, the heat of the pieces would change the solution temperature, so the heat capacity ratio of the heat conducting pieces and the microfluidic chip-solution would affect the ramp rates of the solution. Through thermoelectric equivalence analysis, we can find that after the heat-conducting pieces make contact with the microfluidic chip, their final equilibrium temperature would tend to be [35]:(3)$$T_{f}=\frac{T_{c}+T_{\mathrm{rs}}\left(C_{\mathrm{rs}}/C_{c}\right)}{1+C_{\mathrm{rs}}/C_{c}}$$
where $C_{\mathrm{rs}}$ is the heat capacity of the microfluidic chip solution, and $C_{c}$ is the heat capacity of the heat-conducting pieces; when $C_{\mathrm{rs}}/C_{c}$ tends to zero, that is when the heat capacity ratio between the microfluidic chip solution and the heat-conducting pieces tends to zero, and the final temperature $T_{f}$ of the heat-conducting pieces after making contact with the microfluidic chip tends to an initial $T_{c}$. With $C_{\mathrm{rs}}/C_{c}$ increasing, the difference between $T_{f}$ and $T_{c}$ increases, resulting in a decrease in the ramp rates of the solution. Therefore, the heat capacity of the thermal conductivity pieces relative to the microfluidic chip solution need to be kept large enough.

#### 2.3.3. Determination of Parameters for Heat Generators

During the PCR cycles, the heater needed to generate enough heat with sufficient power, and determining its power and other parameters was crucial. Among the three TZs, because TZ III (the high-temperature TZ) needed to produce the most heat, we selected it to calculate the heater power, and the heater models in other TZs were same, but the corresponding heat was produced by controlling the power. The parameters of the heater were determined according to the theoretical calculation results of heat load, and the material parameters are shown in Table A1.
1.The thermal load consisted of two main components:
(1)The heat absorbed by the microfluidic chip and solution to rise from room temperature (25 °C) to 95 °C:(4)$$Q_{0}=Q_{c}+Q_{r}+Q_{s}=(c_{c}\rho_{c}v_{c}+c_{r}\rho_{r}v_{r}+c_{s}\rho_{s}v_{s})\Delta T_{0}=209\;J$$
where $Q_{c}$, $Q_{r}$, and $Q_{s}$ are the heat absorption of the thermal conductivity piece, microfluidic chip, and solution, respectively.
(2)Heat loss on the bottom and sides of the heat-conducting pieces:The temperature of the heat-conducting pieces was preset to 130 °C, so the thermal physical parameters of the air are shown in Table A2.

The surface heat loss was generated by the natural convection between the vertical sides and the horizontal bottom of the heat-conducting pieces and the air, and the amount of heat loss was
(5)$$Q_{1}=4Q_{\mathrm{cs}}+Q_{\mathrm{cb}}=\left(4h_{\mathrm{cs}}A_{\mathrm{cs}}+h_{\mathrm{cb}}A_{\mathrm{cb}}\right)\Delta T_{1}=1.96\;J$$
where $h_{\mathrm{cs}}$ and $h_{\mathrm{cb}}$ are the convective heat transfer coefficients between the sides and bottom of the heat-conducting pieces and the air, A is the contact area between a heat-conducting piece and the air, and $\Delta T_{1}$ is the temperature difference between the heat-conducting pieces and the surrounding air at 30 °C.

2.Determination of heat generator parameters:

Considering the actual efficiency of the heaters, the selected heaters’ power needed to be greater than or equal to 2 times the average power of the load [36], so:(6)$$P\geq 2P_{a}=2\frac{Q_{0}+Q_{1}}{t}=21.96\;W$$
where *t* = 20 s, i.e., the heater temperature rose from room temperature to 95 °C in about 20 s for pre-denaturation before starting the PCR cycle.

Based on the above calculations and combined with the types of commercially available heaters, the parameters for the selection of heaters are shown in Table 2.

### 2.4. Thermocycler Implementation and Performance Testing

Based on the above design, a prototype of the ultrafast PCR thermocycler was built, as shown in Figure 5. The prototype worked with an external 24 V power supply, and the user could set parameters and sent commands via a computer, as well as export and analyze the run data. The model and performance parameters of the thermocycler components are shown in Table A3.

To test the performance of this thermocycler for liquid thermal cycle control, the following test method was established: a 0.5 mm diameter hole was drilled in the side wall of the microfluidic chip using a CNC, and then the test end of a K-type thermocouple (KAIPUSEN, China) was inserted into the middle of the reaction chamber and fixed with a glue seal, as shown in Figure 6. Finally, a temperature data acquisition instrument (MX100, YOKOGAWA, Tokyo, Japan) was used to monitor the solution temperature in real time (acquisition frequency: 2 Hz). The temperature variation conditions of the solution could be fine-tuned by adjusting the following parameters: the temperature in each TZ, the microfluidic chip dwell time in each TZ, the motor movement speed, and the displacement. Amplification experiments were performed using biological reagents and HCMV samples after each parameter was adjusted to make the solution temperature profile optimal.

In order to make the solution temperature change rapidly between 55 °C and 95 °C and then increase the ramp rates of the solution as much as possible, we used the following strategy: (1) The temperature difference between the TZs was increased. TZ I was set to be much lower than 55 °C, such as 50 °C, 40 °C, or 30 °C, and TZ III was set to be higher than 95 °C (100 °C, 110 °C, or 120 °C). In the process of changing the solution temperature from 95 °C to 55 °C, the microfluidic chip first moved to TZ I for a period of time to release the heat, and to avoid temperature fluctuations in TZ II and avoid affecting the temperature stability. Then, when the solution reached 55 °C, the microfluidic chip immediately moved to TZ II and stayed for a period of time which was determined according to the annealing/extension reference time provided by the reagent. (2) The microfluidic chip moved rapidly during the movement process and attached to the bilateral heat-conducting pieces as soon as possible after reaching each TZ.

### 2.5. Sample and PCR Protocol

To assess the potential of this thermocycler for clinical application, NAA was performed using DNA from HCMV with an amplified fragment of 120 bp [37]. A viral DNA/RNA extraction kit (magnetic bead method, GenMagBio, Changzhou, China) was used to extract the AD169 strain of HCMV virus that was cultured by ARPE19 cells and then stored at −20 °C after dilution of 16 with 1× TE buffer (Sangon Biotech, Shanghai, China). After the DNA was extracted from the virus, it was mixed with reagents to form an amplification system. The amplification system consisted of 3.125 μL of DNA template (35.2 ng/µL), 2.5 μL of buffer, 0.625 μL of 10 mM dNTPs, 0.625 μL of forward primer, 0.625 μL of reverse primer and 0.625 μL of probe, 0.25 μL of taq-enzyme, 7.25 μL of DEPC water, 6.25 μL of special reagent A, and 3.125 μL of reagent B (Beijing Wantai Biological Pharmacy Enterprise, Beijing, China). We used 6 positive samples and 2 negative controls (DNA template was replaced with DEPC water) for NAA in our thermocycler, as well as 3 positive samples for amplification in the Bio-Rad PCR instrument (C1000 Touch^TM^, Bio-Rad, Hercules, CA, USA) and the Tianlong Fast PCR machine (Gentier 96, TIANLONG, Taiyuan, China), respectively, to evaluate the NAA performance of our device. With a pipette, 25 μL of solution was added to the amplification chamber of microfluidic chips, and then 5 μL of paraffin oil was added to block the channels of the microfluidic chips and to prevent the solution from evaporating.

The amplification protocols of the Bio-Rad PCR instrument and the Tianlong Fast PCR instrument were: pre-denaturation at 95 °C for 3 min, denaturation at 95 °C for 5 s, and annealing/extension at 55 °C for 15 s. After amplification, the amplified products were examined with endpoint fluorescence and agar gel electrophoresis. For the endpoint fluorescence method, we examined the fluorescence values before and after amplification using the Bio-Rad PCR instrument including a fluorescence detection system (CFX96^TM^ Real-Time, Bio-Rad, Hercules, CA, USA), and determined whether the amplification was successful by calculating the fluorescence ratio (endpoint fluorescence/initial fluorescence), which theoretically indicates successful NAA as long as the ratio is greater than 1. To prevent any possible occurrence of false positives due to potential errors in the fluorescence signal measurement, we adjusted the fluorescence ratio threshold to 1.07 based on the amplification results of 90 negative controls. The specific measures were: before amplification, each sample was added to 8-strip PCR tubes and then placed into the Bio-Rad PCR instrument for initial fluorescence detection at 30 °C; after each sample was amplified on the corresponding device, the amplification products were transferred to 8-strip PCR tubes and then placed in the Bio-Rad instrument for endpoint fluorescence detection at 30 °C. For the second method, we transferred the above amplified products to an electrophoresis instrument (DYY-7C, Liuyi Biotechnology Co., Ltd., Beijing, China) for electrophoresis and then imaged them to determine the success of amplification according to the imaging effect. Each product was collected separately in a tube and mixed with 1× blue dye; then, 10 μL of sample was added to 1.2% agarose gel and electrophoresed in 10× tricarboxylic acid/boric acid/EDTA (TBE) buffer. The gels were electrophoresed at 200 V for 15 min and then imaged using an imaging system (ChemiDocTMMP Imaging System, Bio-Rad, Hercules, CA, USA).

## 3. Results and Discussion

The following section will show the temperature variation in the solution at different temperature overshoots (temperature differences) in between the TZs, the linear regression models of the solution ramp rates versus the temperature differences, and the thermal performance characterization and NAA testing results of the thermocycler.

### 3.1. Fluid Temperature Changes Driven by Temperature Difference

In order to improve the ramp rates of the solution temperature, we adjusted the linear servo driver to provide good contact between the microfluidic chip and the upper and lower thermal conductors so as to reduce the contact thermal resistance, and adjusted the stepper motor module to make the microfluidic chip move at a speed of 50 mm/s. On this basis, we experimentally studied the ramp rates at different temperature overshoots in TZ I and TZ III and established a linear regression mode.

Based on the demand of PCR, TZ II was set to 55 °C. The temperatures of TZ I and TZ III were changed in turn to make the solution temperature quickly reach and stay at 55 °C for 3 s or quickly reach 95 °C; the test scheme configuration is shown in Figure 7, where the residence time of 55 °C could be set arbitrarily according to the requirements of the biological reagents.

On the one hand, in the heating stage of the solution from 55 °C to 95 °C, TZ I and TZ II were fixed at 50 °C and 55 °C, respectively, and TZ III was set to 100 °C, 110 °C, and 120 °C in turn. The results are shown in Figure 8a and Table 3. The solution heating times were 6.5 s, 4 s, and 2.5 s for 100 °C, 110 °C, and 120 °C, respectively, corresponding to average heating rates of 6.15 °C/s, 10 °C/s, and 16 °C/s, as well as cycle times of 15 s, 13 s, and 11 s. On the other hand, during the cooling stage of the solution from 95 °C to 55 °C, TZ II and TZ III were fixed at 50 °C and 100 °C, respectively, and TZ I was set to 50 °C, 40 °C, and 30 °C in order. The results are shown in Figure 8b and Table 3. It can be seen that during the heating process, when TZ II was kept at 55 °C and TZ III was raised from 100 °C to 120 °C, the heating time was shortened from 6.5 s to 2.5 s, and the heating rate was increased from 6.15 °C/s to 16 °C/s; during the cooling process, when TZ III was kept at 100 °C and TZ I was lowered from 50 °C to 30 °C, the cooling time was shortened from 5.5 s to 3.5 s, and the cooling rate was increased from. Therefore, increasing the temperature difference between the solution and the TZs can, to some extent, raise the solution’s heating and cooling ramp rates and then shorten the PCR time.

To further quantify the relationship between different overshoot temperatures in TZ III and TZ I and the ramp rates of the solution, we conducted experimental tests in multiple temperature conditions. The results showed that the ramp rates of the solution varied linearly with the temperature differences, and the R^2^ value reached above 0.97, indicating an excellent linear fit. In the heating and cooling process, the linear models are y = 0.4699x − 41.02 and y = −0.2158x + 18.20, respectively, where y is the ramp rate, x indicates the temperature of the TZs, the negative number indicates the cooling process, and the larger the absolute value, the larger the temperature difference or cooling ramp rate, as shown in Figure 9.

On the one hand, the linear regression models can be used to predict the heating or cooling ramp rates of the solution at a specific temperature in TZ III or TZ I. As shown in Table A4, the heating and cooling rates of the solution (the predicted values) were calculated using the regression model, and when the predicted values were compared with the experimental values, the average error rates of the heating and cooling rate predictions were only 4.41% and 2.56%, indicating a small prediction error and high accuracy.

On the other hand, the linear regression model can guide the setting of the appropriate temperature difference through the predicted ramp rates of the solution, i.e., setting the temperature of each TZ according to the degree of tolerated temperature change for DNA polymerase in order to achieve fast PCR and not to destroy the biological activity of the polymerase. For example, if the tolerated ramp rate of polymerase is 11 °C/s, the temperature of TZ III and TZ I should be set to 110.7 °C and 33.4 °C, respectively.

### 3.2. Thermal Performance of the PCR Process

Based on the above experiments and considering the tolerance of the used DNA polymerase to rapid temperature change, the temperature control strategy during thermal cycling was set as follows: the temperatures of TZ I, TZ II, and TZ III were set to 30 °C, 55 °C, and 115 °C, respectively, as shown in Figure 10. The residence times of the solution at 95 °C and 55 °C were 0.5 s and 3 s respectively. In addition, TZ III was set to 95 °C during the pre-denaturation stage and the solution dwelt at 95 °C for 1 min.

Meanwhile, to evaluate the thermal cycling performance of the device, two commercial PCR instruments, Tianlong Fast PCR and Bio-Rad PCR, were selected to measure the solution temperature during the PCR process under the same conditions; the actual test results showed that our ultrafast thermocycler could greatly shorten the PCR cycle time, as shown in Figure 11. The maximum heating and cooling ramp rates reached 24.12 °C/s and 25.28 °C/s, respectively, and the average ramp rate was 13.33 °C/s, as shown in Figure 12a, where positive values and negative values indicate the heating rate and cooling rate, respectively. To further evaluate the temperature stability of the annealing/extension stage, all 45 cycles were divided into 3 stages, and the results of each stage are shown in Table A5. The highest and lowest temperature were 56.05 °C and 54.15 °C, respectively, with a maximum error of 1.05 °C, which appeared in the first two cycles because TZ III could not be heated to 115 °C after the end of the pre-denaturation stage, as shown in Figure 12b. The average error of 45 PCR cycles was 0.22 °C, proving good temperature stability.

### 3.3. Nucleic Acid Amplification Results

In the thermocycler with the above debugging procedure, NAA was implemented. The results of fluorescence detection before and after NAA are shown in Table 4; they indicate that the fluorescence value ratio was greater than 1.05 and was similar to that of the amplification products of the Bio-Rad and Tianlong PCR instruments, indicating successful amplification.

The results of the agarose gel electrophoresis assay are shown in Figure 13, where lane M is a marker for different lengths of DNA, which clearly shows the presence of amplification products (lanes P1–P6) with a similar band brightness to the products from the Bio-Rad and Tianlong instruments, while no amplification products were observed in the negative controls (lanes N1 and N2), indicating the absence of primer dimer and the generation of specific amplification. Thus, this thermocycler was able to achieve NAA with 9 min, which is much faster than the 1~3 h required by commercial PCR devices such as the Bio-Rad PCR instrument.

### 3.4. Discussion

From the heat-transfer models in Section 2.3, we know that the key factors affecting the solution ramp rates include: the temperature difference between the TZs and the solution, the thermal resistance, and the heat capacity of the microfluidic chip and solution. Heat transfer is driven by temperature differences, meaning that a greater temperature difference between the solution and heat source will result in faster ramp rates. However, if the temperature difference is too large, it can lead to poor temperature stability, which is not suitable for PCR. Therefore, it is important to balance the temperature difference to achieve optimal ramp rates without compromising temperature stability. In addition, the temperature in TZ III should not be set higher than 120 °C, otherwise it may lead to high temperature in the local solution and then reduce the activity of the taq enzyme. Thermal resistance, including contact and conduction thermal resistance, determines the efficiency of heat transfer; contact thermal resistance is unavoidable, but can be reduced by increasing the compression force and making the reactor chamber membrane bulge; and conduction thermal resistance is reduced by decreasing the thickness of the thin polymer cover layer of the amplification chamber, but very thin membranes (<0.05 mm) are very demanding for bonding (ultrasonic bonding and chemical bonding), so this requires a balance. The efficient heat transfer in our device is due to two aspects: the close contact between the thermal conductivity pieces and the two sides of the amplification chamber, and the thin (0.05 mm) polymer cover layer between the solution and the thermal conductivity pieces. The heat capacity is mainly reflected in the volume of the solution: the smaller the volume, the faster the heating and cooling rates, but a small volume (<10 μL) may not be conducive to subsequent clinical applications. Usually, the clinical volume is greater than 10 μL. The solution volume can be reduced to further increase the thermal cycling rates in our instrument.

The speed of PCR thermal cycling is related to both ramp rates and dwell time. A high ramp rate places high demands on the amplification reagents, especially on the taq enzyme, and dwell time is mainly related to the length of the target DNA fragment; the longer the fragment, the more annealing/extension time is required, even for a three-step PCR. In the NAA of this study, reducing the ramp rates and increasing the dwell time would increase the amplification efficiency, but the total time would increase, which requires a balance. In fact, it is often observed in fast PCR that amplification efficiency or sensitivity is sacrificed in favor of fast results [9]. In many applications, a binary (yes/no) answer is sought whether the target is present or not, especially in clinical diagnostics [38].

Currently, our thermocycler cannot evaluate the efficiency of NAA and quantitative nucleic acid detection or stay for a longer dwell time at 95°C under ultrafast conditions. Therefore, we will design an ultrafast real-time quantitative fluorescence PCR thermocycler (containing five TZs) that can adapt to any PCR thermocycling conditions to solve the current problems in the future.

## 4. Conclusions

In this study, we proposed a temperature control strategy to significantly increase the solution ramp rates by switching microfluidic chips between multiple temperature zones and excessively increasing the temperature difference between temperature zones and the solution; accordingly, we designed an ultrafast thermocycler. This is the first study to introduce temperature differences into rapid thermocycler research, to our knowledge.

The experimental results showed that the ramp rates for the solution temperature are a linear function of temperature differences within a range, and the larger the temperature difference, the faster the rates. Linear regression models can predict the ramp rates for the solution at a specific temperature in temperature zone I and temperature zone III and can also guide the setting of the temperature of each temperature zone according to the predicted rates. Based on the experimental test results and the prediction results of the linear regression models, the three temperature zones were set to 30 °C, 55 °C, and 115 °C, respectively. Using this strategy, the maximum heating and cooling rates for the 25 μL solution reached 24.12 °C/s and 25.28 °C/s, respectively; the average rate was 13.33 °C/s, which is 6~8 times that of conventional commercial PCR devices, and the average error of the annealing/extension stage was only 0.22 °C, indicating accurate temperature control. DNA from HCMV was used for nucleic acid amplification with 1 min pre-denaturation and 45 temperature cycles, and both endpoint fluorometry and agarose gel electrophoresis detection achieved successful ultrafast nucleic acid amplification in 9 min, significantly reducing the PCR time and having value for rapid clinical nucleic acid detection.

## Figures and Tables

**Figure 1 micromachines-14-00658-f001:**
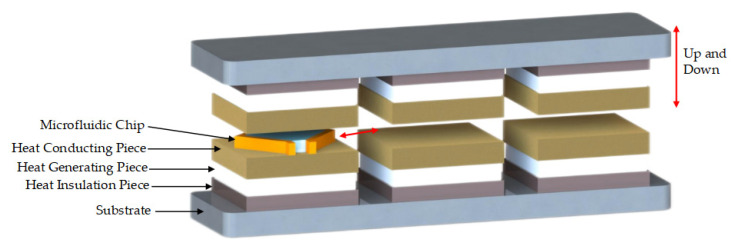
Ultrafast thermocycler scheme.

**Figure 2 micromachines-14-00658-f002:**
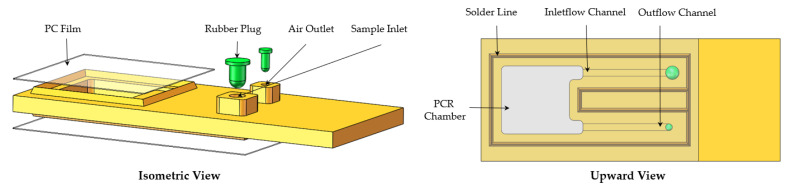
Labeled schematic of the microfluidic chip.

**Figure 3 micromachines-14-00658-f003:**
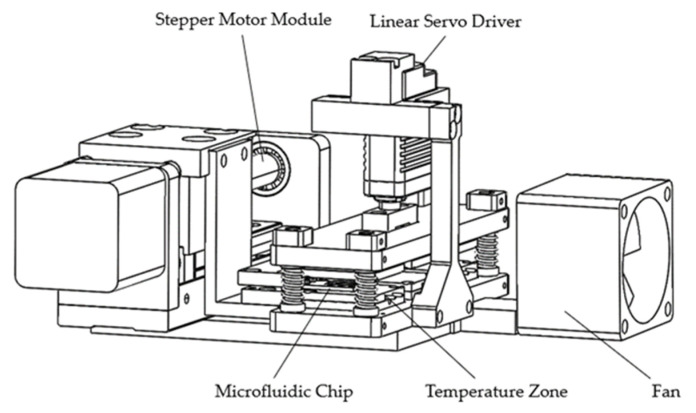
Three-dimensional diagram.

**Figure 4 micromachines-14-00658-f004:**
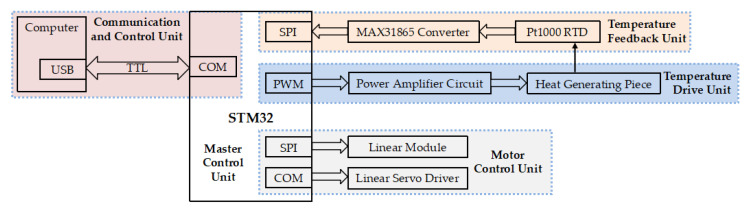
Block diagram of the control unit.

**Figure 5 micromachines-14-00658-f005:**
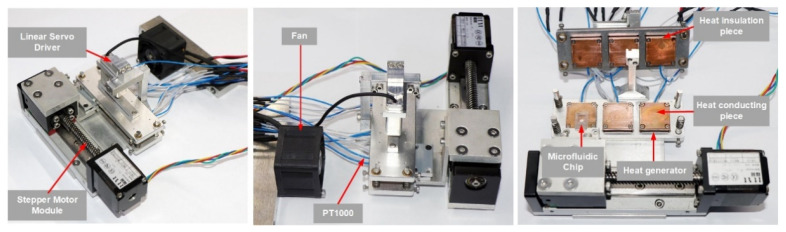
Thermocycler prototype.

**Figure 6 micromachines-14-00658-f006:**
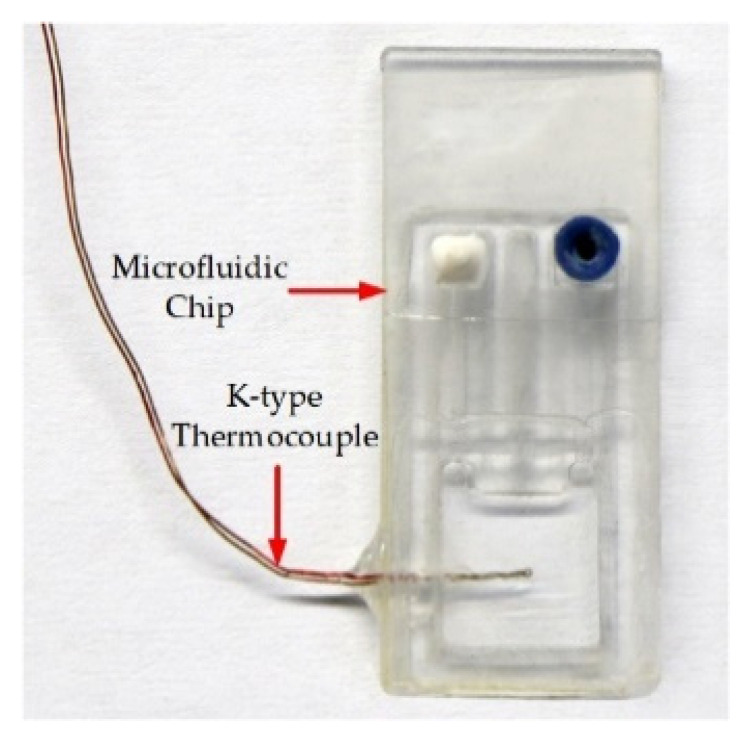
Solution temperature monitoring tooling.

**Figure 7 micromachines-14-00658-f007:**
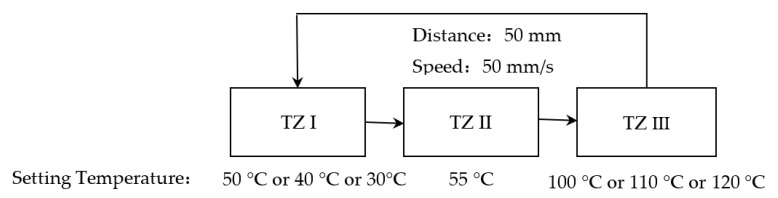
Different physical configurations to study the effect of different temperature overshoots on the ramp rates of the solution.

**Figure 8 micromachines-14-00658-f008:**
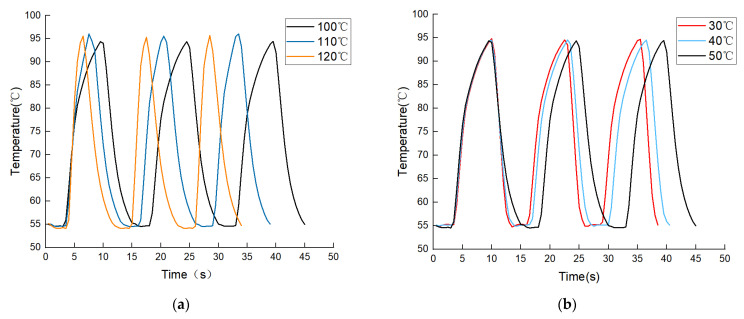
Temperature variation in three PCR cycles of the solution at different temperature overshoots for TZ III and TZ I: (**a**) 3 PCR cycles of TZ III at 100 °C, 110 °C, and 120 °C. (**b**) Three PCR cycles of TZ I at 50 °C, 40 °C, and 30 °C.

**Figure 9 micromachines-14-00658-f009:**
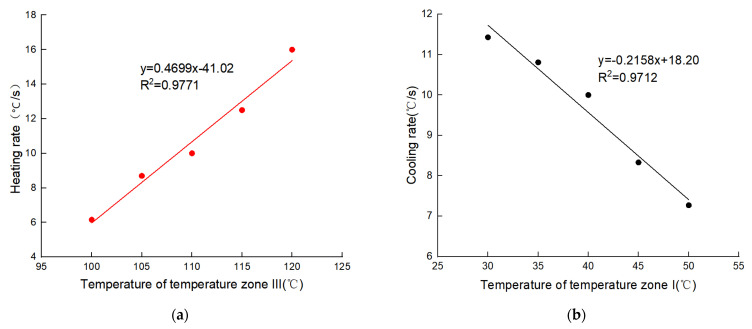
The ramp rate of solution at different temperatures (different temperature differences) in TZ III and TZ I: (**a**) Linear regression model of heating ramp rate and temperature at different temperatures in TZ III; (**b**) Linear regression model of cooling ramp rate and temperature at different temperatures in TZ I.

**Figure 10 micromachines-14-00658-f010:**
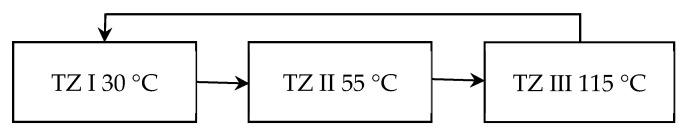
Physical configuration of the PCR process.

**Figure 11 micromachines-14-00658-f011:**
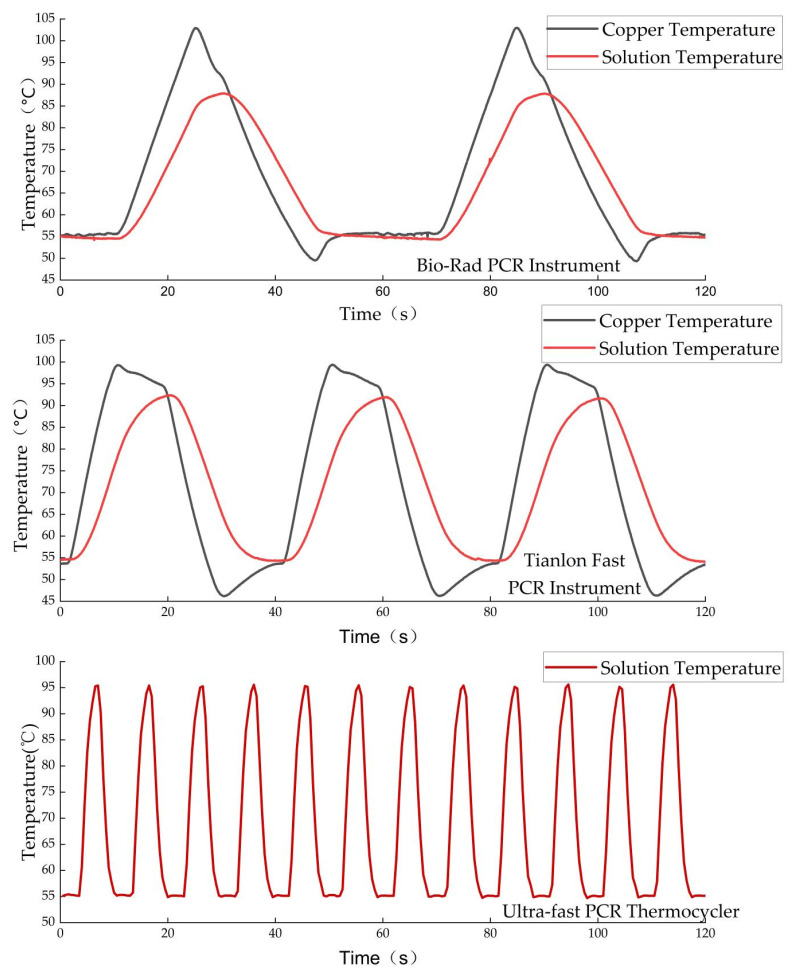
Comparison of solution temperatures between 55 °C and 95 °C for three PCR devices.

**Figure 12 micromachines-14-00658-f012:**
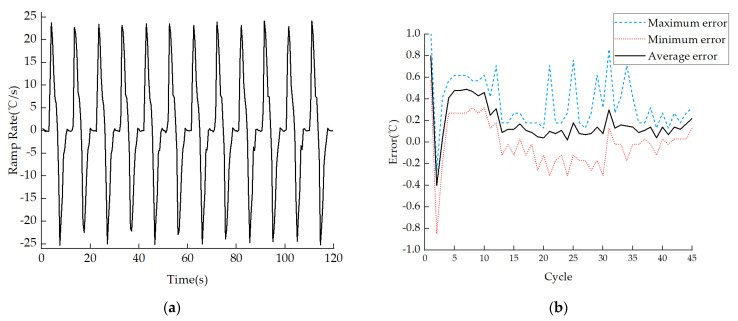
Ultrafast PCR thermocycler thermal cycling performance: (**a**) Heating and cooling ramp rates of solution; (**b**) Error between actual temperature and target temperature of solution in annealing stage.

**Figure 13 micromachines-14-00658-f013:**
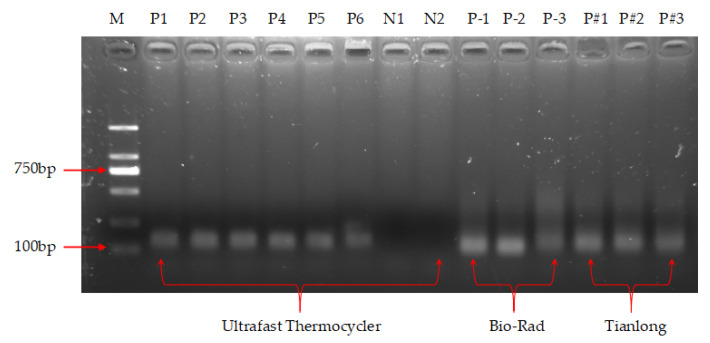
Agarose gel electrophoresis results of nucleic acid amplification. Lane M: DNA markers of different lengths; lanes P1, P2, P3, P4, P5, and P6: positive control of the thermocycler; lane N1 and N2: negative control of the thermocycler; lanes P–1, P–2, and P–3: products of the Bio-Rad instrument; lanes P#1, P#2, and P#3: products of the Tianlong instrument.

**Table 1 micromachines-14-00658-t001:** Comparison of rapid PCR instruments.

Group	Sample Volume	Heating Rate	Cooling Rate	Heating/Cooling Methods	PCR Type	Reaction Monitoring
Qiu et al. [8]	40 μL	Total time: 30 min (Fluorescence intensity saturation)		Resistive heater	Convection PCR	Fluorescence detection
Farrar et al. [9]	1–5 μL	26–130 °C /s		Water bath heating/cooling	Static PCR	Gel electrophoresis
Neuzil et al. [10]	0.1 μL	175 °C/s	125 °C/s	Film heater/natural cooling		Fluorescence detection
Cheong et al. [11]	10 μL	13.17 °C/s	4.94 °C/s	Lasers and Au particles heating/natural cooling		Gel electrophoresis and fluorescence detection
Son et al. [12]	10 μL	12.79 °C/s	6.6 °C/s			Gel electrophoresis
Our previous work [13]	80 μL	4.5 °C/s	5 °C/s	Peltier/fan		Fluorescence detection
Several other researchers [14,15,16]	<10 μL	3–6 s/cycle		Peltier, resistive heater, or film heater	Continuous flow PCR	Gel electrophoresis or fluorescence detection

**Table 2 micromachines-14-00658-t002:** Heater parameters.

Voltage V	Power P	Dry Firing Surface Temperature
24 V	20 W	300 °C

**Table 3 micromachines-14-00658-t003:** Thermal performance of solution at different temperature overshoots in TZ III and TZ I.

**Heating Process**	**Temperature of TZ II**	**Temperature of TZ III**	**Heating Time**	**Average Heating Ramp Rate**	**Single PCR Cycle Time**
	55 °C	100 °C	6.5 s	6.15 °C/s	15 s
		110 °C	4 s	10 °C/s	13 s
		120 °C	2.5 s	16 °C/s	11 s
**Cooling Process**	**Temperature of TZ III**	**Temperature of TZ I**	**Cooling Time**	**Average Cooling Ramp Rate**	**Single PCR Cycle Time**
	100 °C	50 °C	5.5 s	7.27 °C/s	15 s
		40 °C	4 s	10 °C/s	13.5 s
		30 °C	3.5 s	11.43 °C/s	12.5 s

**Table 4 micromachines-14-00658-t004:** Fluorescence detection results before and after nucleic acid amplification.

	Sample Label	Initial Fluorescence Value (RFU)	End Fluorescence Value (RFU)	Fluorescence Ratio
Ultrafast Thermocycler	P1	3965	4708	1.19
	P2	3960	4433	1.12
	P3	4032	5211	1.29
	P4	3912	5125	1.31
	P5	4160	5359	1.29
	P6	3938	4713	1.20
	N1	4322	3574	0.83
	N2	4273	3443	0.81
Bio-Rad Instrument	P-1	4083	5530	1.35
	P-2	4080	5741	1.41
	P-3	4164	6129	1.47
Tianlong Instrument	P#1	4106	5514	1.34
	P#2	4124	5289	1.28
	P#3	4068	5297	1.30

## Data Availability

Not applicable.

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
