# Peer review of "Ultrafast Microfluidic PCR Thermocycler for Nucleic Acid Amplification"

_micromachines, 2023, doi:10.3390/mi14030658_

Round 1
Reviewer 1 Report
See Word file

Reviewer 2 Report
1, The thermal cycle should include an temperature holding time. It can not be found in the test. Please explain this.
2, The fluorescence test should be performed by the proposed fast method and the commercial methods of Bio-rad and Tianlong instrument. The amplification result is important except the process time.
3, Table 6 should provide unit of the fluorescence value. otherwise, it is difficult to evaluate the real result.
Reviewer 3 Report
In this manuscript, authors proposed a temperature control strategy and accordingly designed an ultra-fast PCR nucleic acid amplification system. Research data can support the conclusion. However, there are still some problems. I think this work is not innovative enough to be published in micromachines.
(1) The biggest problem of the ultra-fast PCR nucleic acid amplification system proposed in this paper is that the efficiency of amplification could not be evaluated and quantitative nucleic acid detection could not be performed due to the lack of corresponding real-time fluorescence detection system, so it has no application value.
(2) What are the advantages of this device compared to other rapid amplification devices developed to transfer PCR tubes? Since instruments such as the BIO-RAD do not require moving PCR tubes, and the principle of temperature increasing and decreasing process is different from that of the device, so the comparison is not very meaningful.
(3) In this paper, the results were determined by the endpoint method and gel electrophoresis characterization, which requires data calculation, and there will be errors affecting the accuracy of the results and the operation is cumbersome, which is a very big problem.
(4) The sensitivity of the assay cannot be guaranteed by this device for amplification assays, which are not supported by relevant data, and the results calculated by the endpoint method will further affect the sensitivity of the detection assay.
Reviewer 4 Report
In this manuscript, the authors designed different temperature regions for different PCR amplification temperatures, and performed equation derivation and theoretical calculations. Finally, the usefulness of the system was verified by the amplification of HCMV DNA. However, the article is not very innovative, especially in the part of validating the usability of the system using clinical samples. Not only the comparison with conventional PCR amplification results is missing, but also the validation with actual clinical samples is not available. There are a lot of repetitive statements in the manuscript, such as lines 319 to 329 and lines 457 to 463.
A lot of ultrafast and specific PCR technologies are also proposed now, such as Minli You et al., Trends in biotechnology 2020, 38 (6), 637-649 ; Son, J et al. Light Sci Appl 4, e280 (2015) and Sang HunLee et al., Biosensors and Bioelectronics 2019, 141, 111448. The method of ultrafast PCR proposed by the authors here does not seem to have advantages over these methods mentioned above.
Round 2
Reviewer 1 Report
1. The current/adapted title is not very clear.
2. The table is an improvement, but the text can be shortened a bit due to this table.
3. If two step PCR can be used is not (only) dependent on the lenght of the template. Please adapt and use the correct arguments!
4. What is 28Ct for concentration? Please report in ng/µL or cells/µL or a similar unit.
5. The increase in fluorescent signal and the fluorescence ratio are quite small. What about significance? And why is the ratio below 1.0 for negative controls? It this always the case. What is the threshol to say a sample is negative or positive?
6. The (clinical) sensitivity has nothing to do with the 'window period'. It is defined in other terms!
Reviewer 4 Report
The authors have added a comparison of the results of nucleic acid amplification by different instruments to the revised manuscript and have made extensive changes to repetitive passages and the language of the article. The logic of the study design has been improved and the innovation is clearer, so I agree to accept it.
Author Response
We appreciate your approval of our work. We would like to take this opportunity to express our gratitude to you for your valuable feedback and insightful suggestions which have helped us improve the manuscript significantly. We wish you all the best in your future endeavors and personal lives.
Round 3
Reviewer 1 Report
While the manuscript is improved a lot, there are still some substantial issues to be addressed:
1. The title of the manuscript is still a bit weird concerning English grammar. Especially the 'Temperature Difference Driving' part.
2. I would recommend to check the article by a native English speaker. There are more sentences that can be improved.
3. I still miss the fact that being able to perform 2-step PCR, the annealing temperature of the primers should be suitable in order to combine it with the extention temperature.
4. I still miss a statistic justification for the fluorescent ratio to be set on 1.05 as threshold.
Round 4
Reviewer 1 Report
There is still a 1.05 value in the manuscript, should that also be changed to 1.07?